# Comparing Anxiety Levels during the COVID-19 Pandemic among Mothers of Children with and without Neurodevelopmental Disorders

**DOI:** 10.3390/children10081292

**Published:** 2023-07-27

**Authors:** Ivana Bogavac, Ljiljana Jeličić, Jelena Đorđević, Ivana Veselinović, Maša Marisavljević, Miško Subotić

**Affiliations:** 1Cognitive Neuroscience Department, Research and Development Institute “Life Activities Advancement Institute”, 11000 Belgrade, Serbia; i.bogavac@add-for-life.com (I.B.); m.marisavljevic@add-for-life.com (M.M.); m.subotic@add-for-life.com (M.S.); 2Department of Speech, Language and Hearing Sciences, Institute for Experimental Phonetic and Speech Pathology, 11000 Belgrade, Serbia; 3Department of Psychiatry, Faculty of Medical Sciences, University of Kragujevac, 34000 Kragujevac, Serbia; jelena.djordjevic@medf.kg.ac.rs; 4Clinic for Neurology and Psychiatry for Children and Adolescents, 11000 Belgrade, Serbia; 5Department of Defectology-Hearing Disability, Faculty of Special Education and Rehabilitation, University of Belgrade, 11000 Belgrade, Serbia; ivanaveselinovic@fasper.bg.ac.rs

**Keywords:** neurodevelopmental disorders, autism spectrum disorder, specific language impairment, anxiety, parents, children, COVID-19

## Abstract

The COVID-19 pandemic undoubtedly burdened families, perhaps even more for parents of children with neurodevelopmental disabilities. This research aims to determine the anxiety levels in mothers of children with neurodevelopmental disorders (autism spectrum disorder and specific language impairment) and mothers of typically developed children. The cross-sectional study comprised 280 mothers from the period of the COVID-19 pandemic in Serbia. A confidential survey included main demographic data and the State-Trait Anxiety Inventory (STAI). Results revealed that the mean levels of STAI-S and STAI-T are elevated in the observed sample of mothers in the first pandemic wave; the STAI-S level is in the high category (STAI-S mean = 46.69), while STAI-T is in the intermediate category near the cut-off value for the high level (STAI-T mean = 43.04). A statistically significant strong positive correlation between STAI-S and STAI-T is seen (r = 0.802, *p* = 0.001). GLMM analysis revealed that interactions, rather than independent variables, significantly impact anxiety, implying a complex relationship between the observed variables and STAI. Compared with the results from the pre-pandemic study, our findings reveal that COVID-19 affects mothers of children with and without neurodevelopmental disorders in a complex manner, imposing a need for psychological support, which may positively affect mothers’ mental health and the development of their offspring.

## 1. Introduction

The World Health Organization (WHO) announced a worldwide epidemic for respiratory disease caused by the SARS-CoV-2 coronavirus in March 2020 [1]. Due to extreme virulence, governments worldwide have implemented different prevention measures to stop the spread of the virus. Such measures included mandatory gloves, protective masks and eyewear, maintaining social distance, canceling social events, travel restrictions, and lockdowns with strict isolation measures. After China and Italy, at the beginning of March, other governments introduced lockdowns as a measure: schools, universities, the majority of workplaces (except healthcare, press, and suppliers of primary assets and food) were closed, and people stayed in their homes. All supportive educational, sports or entertainment services directed at children were closed. In Serbia, a state of crisis throughout the country was declared on 15 March 2020 [2]. Domestic lockdown and social isolation were very effective in terms of protecting physical health and preventing the further spread of the virus. However, these strict, necessary measures could have adverse consequences for the mental and physical well-being of individuals, especially children [3] and their parents [3,4]. This was a period of uncertainty with no optimism that it would end soon. Parents and their children had to adjust to these new and complex living, working, and learning conditions. Imposed isolation can cause psychological problems such as anxiety, depression, insomnia, or stress symptoms [5]. This was a period of the undermining of mental health, with the restricted activity of mental institutions and with worsening symptoms in pre-existing patients and emerging symptoms in individuals without a positive history of mental problems [6]. Some studies have reported psychological problems and increased stress among parents and caregivers during the COVID-19 pandemic [7,8,9,10,11], while others have focused on vulnerable groups of parents with developmentally disabled children [12,13,14,15].

The term neurodevelopmental includes a very broad group of mental disorders [16] characterized by developmental deficits in neurological and psychiatric functioning, which can be lifelong conditions. In this paper, we will only discuss the aspects of autism spectrum disorder (ASD) and speech and language impairment (SLI) as a part of this complex group of conditions. ASD represents a spectrum of different manifestations that include a lack of social interaction, language impairment, and behavioral disorders (restricted and repetitive interests), as well as a decreased or increased sensory sensitivity, problems with attention, hyperactivity, minor or major disturbances in sleep, and emotional and/or mood regulation [17,18,19]. SLI refers to difficulties in acquiring and the use of language (spoken, written, or other) as a consequence of comprehension or production difficulties [20] in children with an absence of other developmental delays [21]. As mentioned above, SLI symptoms persist in children with ASD and are an example of comorbid disorders, which can share the symptomatology and possible outcomes [18,20]. Both conditions require early identification and intervention [22,23]. The treatment dynamic is individualized according to every child’s needs and recommendations. Every child has their own dynamic in developing new functions and implementing them in everyday functioning, and speech and language therapists adjust their plans and programs accordingly on a daily basis. The focus and aim of therapy is to develop better auditory perception, enrich passive and active vocabulary, encourage proper emotional response and reaction, and raise motivation for interaction and spontaneous communication. The integration of parents and their active role is essential in the therapy process [24,25]. A treatment with an effective structure, which is frequent and long-term, is the most effective at decreasing symptomatology and levels of disability [26,27]. The limitation and restriction of treatments are expected to result in elevated levels of stress after observing the benefits that children gain from treatment and facilitation in everyday life for the whole family.

As mentioned, the pandemic-imposed quarantine restricted all social contact and interaction, and consequently, significant methods of stress relief became unavailable. Studies of previous infection outbreaks have shown significant psychological distress [28,29,30,31]. The evidence for COVID-19 has shown similar trajectories [32,33,34], especially in the parents’ group [11,35,36]. Social support impacts stressful life periods, serving as a buffer and providing emotional, logistical, and practical physical support by family members, neighbors, colleagues, and friends [37]. During the COVID-19 pandemic in Serbia, the lack of social support and the sense of helplessness among the parents of children with autism spectrum disorder resulted in higher stress levels [38].

Stress significantly increases among parents of children with neurodevelopmental disorders and adversely affects parental behavior, their relationship with the child, and the parents’ general health [39]. The parenting stress level throughout the COVID-19 pandemic was higher compared to the period before COVID-19 in both parents with and without children with developmental disorders. However, the overall stress was significantly higher in parents of children with developmental disabilities [40]. However, parents of children with neurodevelopmental disorders faced more adversities and placed more effort into caring for children’s everyday activities, such as learning, individualized treatment, or training, than parents of typically developing children [4]. All of this can cause subjective feelings such as stress, tension, worry, and nervousness, increasing the already generalized tendency of parents toward anxiety. 

Parent anxiety is usually related to parenting stress [41]. Past studies have reported that mothers of children with developmental disabilities had higher anxiety and stress levels than fathers [42,43] and pointed to a positive link with maternal anxiety [44]. One more interpretation is that the mother is the one who is more involved in the children’s daily routine, which may influence their anxiety levels [45]. Studies of anxiety include state and trait anxiety as complementary concepts. State anxiety represents physiological and psychological manifestations directly connected to specific circumstances. Trait anxiety represents a personality trait, a person’s ability to present state anxiety, and therefore is stable over time [46]. During the COVID-19 pandemic, increased anxiety could have resulted from the fact that restrictions in the form of lockdown, as well as online education, placed more burden on children with disabilities and, consequently, on their parents or caregivers [47,48].

To our knowledge, no research has investigated the impact of COVID-19 on the psychological aspects, particularly anxiety levels, of mothers of children with and without neurodevelopmental disorders throughout the COVID-19 pandemic in Serbia. Bearing on the fact that parents are a sensitive population, especially the parents of children with developmental disabilities, this study aimed to compare self-reported anxiety levels among mothers of children with neurodevelopmental disorders (autism spectrum disorder (ASD) and specific language impairment (SLI)) and mothers of typically developed children, in the context of the COVID-19 pandemic in Serbia. Mothers’ demographic characteristics were controlled in the study. We also aimed to identify how the number of children in a family is associated with anxiety levels. 

## 2. Materials and Methods

### 2.1. Study Design and Participants

A cross-sectional investigation of anxiety levels throughout the COVID-19 pandemic among mothers of children with and without developmental disabilities was conducted to determine if and to what extent the pandemic impacts maternal anxiety. The study sample initially included parents in Serbia, recruited online via social media. All parents were invited to complete a confidential online survey. They were briefed on the study’s purpose: to compare anxiety levels in parents of children with and without neurodevelopmental disabilities during the pandemic. Participants did not receive monetary compensation for participating in the survey; it was voluntary. The initial part of the survey addressed demographic data like gender, age, region, educational background, marital status, number of children in the family, and general information regarding parents with children with developmental disorders. All the data were collected throughout the first wave of the COVID-19 pandemic, characterized by the most extreme lockdown (from April to June 2020) in Serbia. After distributing the survey via social networks, 327 responses were received, with valid information on all variables in 319 participants, which is a 97.55% valid rate. The majority of participants who completed the survey were mothers, 305 (M_age_ = 38.72, SD = 5.85, range = 23–54), accounting for 95.6%, and only 14 fathers (F_age_ = 41.17, SD = 5.79, range = 36–57), accounting for 4.39%. Considering the minimal number of fathers, the excluded criteria were used, and the final sample included only mothers.

Regarding sociodemographic variables, it is critical to highlight a significant disproportion between respondents with average, low, and high household incomes in comparison to the small number of respondents with low or high incomes. As a result, average household income became one of the inclusion criteria. Also, a significant disproportion was noticed concerning marital status; an extensive number of participants were co-habiting/married, and a small number of participants were single or divorced. Accordingly, it resulted in co-habiting/marriage being an inclusion criterion. Finally, most children with neurodevelopmental disorders had no interruption in the rehabilitation process, and only a small number did, which is why the inclusion criterion was imposed for including within the study only those children who had been receiving treatment. The final sample comprised 280 mothers of children with and without neurodevelopmental disabilities who were assessed for anxiety throughout the COVID-19 pandemic in Serbia. The survey contained a confidential statement indicating anonymity, data processing, and results dissemination. Respondents were given detailed information about the research. They provided written consent once they accepted to participate in the study.

### 2.2. Measures

The parent’s anxiety levels were assessed via the State-Trait Anxiety Inventory (STAI) [49] as the most commonly used self-report [50]. STAI was used for two segments. First, the state anxiety scale (STAI-S) measures transient anxiety states under pandemic conditions through questions that include subjective feelings of tension, worry, nervousness, and trepidation at the exact moment (state anxiety). In that way, it can be concluded that the actual intensity of anxiety is a consequence of stressful procedures or situations [51]. Second, the trait anxiety scale (STAI-T) measures relatively stable segments of anxiety tendency and general conditions of tranquility, faith, and security [52]. Both scales have 20 questions each, and every question was answered on a 4-point Likert scale (“almost never”, “sometimes”, “often”, and “almost always” for the trait anxiety, and from “not at all”, “somewhat”, “moderately so”, and “very much so” for the state anxiety). The overall result moves from 20 to 80 for each scale, and a higher result points to higher levels of anxiety. The scores are classified into three major groups: scores from 20 to 30 points are of the “no or low anxiety” group, scores from 31 to 44 points are of the “moderate anxiety” group, and scores from 45 to 80 points are of the “high anxiety” group. In this study, we used the same classification for the reasons described in the literature [53,54].

### 2.3. Statistical Analysis

The Statistical Package for the Social Sciences version 22.0 was used for statistical data analysis. First, descriptive analysis was carried out to determine the central tendencies and distributions of the variables. In order to establish if there is a difference between variables of interest, one-way ANOVA was used. We first checked the equality of error variances using Levene’s test, and to determine if equality was not violated in a post hoc analysis, the Bonferroni test for multiple comparisons was used. Otherwise, we used the Games–Howell test for multiple comparisons.

A generalized linear mixed model (GLMM) was used to test the dependences of STAI-S and STAI-T on the mother’s education level, presence/absence of disabilities, the number of children in the family as main effects, and the mother’s age as a covariate. Interactions between main effects were included in the model. Model options were: probability function normal; link function identity; number of iterations = 400; 95% confidence level; degrees of freedom varied across tests (Satterthwaite approximation); and tests of fixed effects and coefficients—using robust estimation to handle violations of model assumptions (robust covariance).

Before analysis, new variables for STAI-S level and STAI-T level were defined. For both, the main variables were divided into three groups with rage limits: if STAI-S/STAI-T was ≤30, the level was low; if STAI-S/STAI-T was between 31 and 44, the level was intermediate; and if values were ≥45, the level was high.

## 3. Results

### 3.1. Sample Characteristics

The study’s sample included 280 participants with a mean age of 38.35 (SD = 5.52). The mother’s age, education level, number of children in the family, presence/absence of disorders, types of neurodevelopmental disorders, and anxiety levels for both STAI scales are shown in Table 1.

Table 2 displays descriptive statistics on anxiety levels. Participants’ anxiety levels ranged from low to high on the STAI-S or STAI-T scales. The mean value for STAI-S and STAI-T of the total sample showed a high level and an intermediate close to the cut-off point for high level (46.69 and 43.04 respectively, the cut-off point is 45).

### 3.2. Correlation of Maternal Age, STAI-S, STAI-T

A correlation analysis between maternal age, STAI-S, and STAI-T was conducted for all the four groups related to mothers’ education level from the whole sample. Before correlation analysis, we checked for a statistically significant age difference between those four groups. One-way ANOVA determined no statistically significant age difference (F(3.276) = 1.589, *p* = 0.192). Correlation analysis between variables showed a strong [55] positive statistically significant correlation between STAI-S and STAI-T (Table 3). We found a weak positive statistically significant correlation between mothers’ ages and STAI-S and STAI-T only in mothers with an education level of 14 to 16 years.

### 3.3. GLMM Analysis

GLMM model for STAI-S revealed that *the mothers’ age*, the two-way interaction term *mothers’ education level* * *presence/absence of disorders,* and the three-way interaction term *mothers’ education level* * *number of children in the family* * *presence/absence of disorders* have a statistically significant impact on the model (Table 4).

The GLMM-estimated STAI-S means of the three-way interaction term *mothers’ education levels* * *number of children* * *presence/absence of disorders* are presented in Figure 1.

The GLMM model with the same variables and parameters was applied to the STAI-T. According to the model, the two-way interaction *mothers’ education level* * *presence/absence of disorders*, *number of children in the family* * *presence/absence of disorders* and the three-way interaction *mothers’ education level* * *number of children in the family* * *presence/absence of disorders* have a statistically significant impact (Table 5).

None of the observed independent variables had a statistically significant influence on the models for STAI-S and STAI-T as the main effect. However, statistically significant influences occurred through mutual two- and three-way interactions. In both models, there is a three-way interaction term, *mothers’ education level* * *number of children in the family* * *presence/absence of disorders*, which, with statistical significance, influences the prediction of STAI.

Figure 2 shows the nonlinear interactions among three observed factors.

Because the interaction term *mothers’ education level* * *number of children in the family × presence/absence of disorders* has 16 subgroups, we analyzed separate groups based on *mothers’ education level*. A new grouping variable was composed for the interaction term *presence/absence of disorders* * *number of children* (Table 6) in order to determine if there is a statistically significant difference between the subgroups in each education group.

For the new grouping variable, one-way ANOVA was conducted. The assumption of the equality of variances was checked using Levene’s test of equality of error variances (Table A1).

Results for the one-way ANOVA test are given in Table A2.

If a statistically significant difference exists, a post hoc test is conducted, and the results of the subgroups’ comparison are presented in Table A3.

One-way ANOVA showed a statistically significant difference between some subgroups within different mother groups based on education level. In the groups of mothers whose education level was up to 12 years and from 12 to 14 years, the post hoc test revealed a statistically significant difference between subgroups only for the STAI-T. In the group of mothers whose education level was from 14 to 16 years, the post hoc test revealed a statistically significant difference between subgroups for the STAI-S and the STAI-T. In the group of mothers whose education level was 12 years, there was a statistically significant difference in STAI-T between mothers of one child without a disorder, mothers of one child with a disorder, and mothers of two or more children without a disorder. Mothers of one child without a disorder had higher levels of STAI-T. In the same educational group, mothers with two or more children whereby one child had a disorder have a higher statistically significant STAI-T than mothers of one child with a disorder and mothers of two or more children without a disorder. In the group of mothers whose education level was between 12 and 14 years, there was a statistically significant difference in STAI-T between mothers of one child without a disorder and mothers of one child with a disorder, and mothers of two or more children having a child with a disorder. Mothers of one child without a disorder have higher levels of STAI-T. In the group of mothers whose education level was between 14 and 16 years, there was a statistically significant difference in STAI-T and STAI-S between the subgroups of mothers. In both cases, mothers of one child with a disorder have higher statistically significant STAI-S and STAI-T than mothers of one child without a disorder and mothers with two or more children without a disorder. Figure 3 presented STAI-S and STAI-T mean values for different education level groups. A statistically significant difference between subgroups of grouping variable based on the interaction term *number of children in the family* * *presence/absence of disorders* was found.

## 4. Discussion

The adverse effects of the SARS-CoV-2 coronavirus have been analyzed in many studies in the past three years, considering humans’ physical and mental health. Different studies investigated the influence of the virus on different populations from various points of view: physiological, psychological, social, and economic [56,57,58,59]. A sensitive population, the population of parents, has been the focus of different researchers [35,60,61], and some have focused on the most sensitive group of parents and children with developmental disabilities [62,63,64,65,66,67]. Only one study in Serbia has investigated the challenges of raising ASD children throughout the COVID-19 pandemic [38]. Our study is the first known study in Serbia to evaluate the impact of COVID-19 on psychological aspects and anxiety levels, particularly of the mothers of children with and without neurodevelopmental disorders.

The present cross-sectional study compared self-reported anxiety levels in mothers of children with and without neurodevelopmental disorders during the most extreme COVID-19 lockdown in Serbia. We focused on mothers due to the significantly small number of fathers participating in our study (4.39%). Our aim was also supported by the indication from the literature that mothers more often take care of children [68,69,70] and that mothers in Serbia are more stay-at-home parents [38].

The main findings of our study were: the mean levels of STAI-S and STAI-T are elevated in the observed sample in the first pandemic wave; STAI-S level is in the high category, while STAI-T is in the intermediate category near the cut-off value for a high level; a statistically significant strong positive correlation between STAI-S and STAI-T is obtained; and GLMM analysis revealed that interactions rather than independent variables have a significant impact on anxiety. It implies a complex relationship between observed variables and STAI.

### 4.1. Anxiety Levels throughout the COVID-19 Pandemic among Mothers of Children with and without Neurodevelopmental Disorders 

The study revealed a small number of mothers with low levels of anxiety identified using STAI-S and STAI-T throughout the COVID-19 pandemic in Serbia. The mean value of STAI-S for the whole sample is in the high category (46.69), and it is higher than the mean value of STAI-T (43.04), which is in the intermediate category, very close to the cut-off point for the high category. STAI-S and STAI-T revealed high anxiety levels in 60.4% and 38.6% of mothers, respectively. Several recent studies have found an increase in anxiety symptoms in parents and caregivers throughout the COVID-19 pandemic, consistent with our findings [9,11,12,13,14,15,35]. This trend is also observed in children with developmental disabilities [63,65,71]. The increased anxiety levels throughout the COVID-19 pandemic were also reported in the general population [72], as well as the trend of higher anxiety levels at the beginning of the pandemic [73,74]. It is important to emphasize that psychological distress data may be particular to one country due to various lockdown measures implemented during the pandemic [75]. Responders from Serbia have shown a higher resilience to COVID-19 in the general population than responders from Italy, Lebanon, and Portugal, presumably due to previous sociocultural experiences [76]. On the other hand, our findings on anxiety levels in the mothers of children with and without neurodevelopmental disorders throughout the COVID-19 pandemic in Serbia are consistent with the research mentioned above, indicating higher anxiety levels throughout the COVID-19 pandemic [63,65,71].

Considering findings about parental stress and anxiety before the COVID-19 pandemic, researchers mainly focused on the parents of ASD children. They noted that they experienced more parenting stress [77,78,79,80] and excessive anxiety [81,82] than parents of typically developing children [79] or parents of children with other developmental disabilities [83,84]. The elevated parental stress and anxiety levels may be explained by the child’s inappropriate and unpredictable behaviors, the severity of symptoms, the child’s emotional problems, the feeling of a lack of support by their environment, and financial worries secondary to the need to spend for treatment intervention and education [80,84]. Based on such findings before the COVID-19 pandemic, the phenomenon of an increase in anxiety symptoms in the parents and caregivers of children with neurodevelopmental disorders during the COVID-19 pandemic was to be expected [63,65,71]. In that sense, recent research throughout the COVID-19 pandemic is consistent with our study findings, which pointed to the elevated levels of anxiety symptoms as well as parenting stress, and depressive symptoms among the parents of children with various developmental disorders such as language and speech disorders, learning disabilities, attention-deficit/hyperactivity disorder, and ASD during the pandemic [13].

### 4.2. Correlation between Maternal Age, STAI-S, and STAI-T

A correlation analysis between maternal age, STAI-S, and STAI-T showed a strong positive statistically significant correlation for STAI-S and STAI-T. This applies to the entire sample and groups of mothers with different levels of education. No correlation was found for maternal age and STAI-S, as well as for maternal age and STAI-T when the whole sample was observed. A weak correlation was found for maternal age and STAI-S, as well as for maternal age and STAI-T in the group of mothers whose education level was from 14 to 16 years. The literature data from the population reveals ambiguous results about the correlation between STAI-S and STAI-T; and age, STAI-S, and STAI-T. The findings cover a wide range from no correlation to a strong correlation in both directions, positive or negative [85,86,87,88,89,90]. Such a wide dispersion between the obtained results can be related to various experimental conditions, including different anxiety traits, sociodemographic factors, sociocultural aspects, or situations related to health policy [91]. Our and other research results indicate that it is difficult to make conclusions about the correlation even if we want to generalize from population subgroups’ results or drive conclusions from the general population to a specific subgroup of interest.

In addition, the high statistically significant correlation between STAI-T and STAI-S could be interpreted as a result that supports the theoretical assumption that anxiety is a unidimensional construct [46]. However, such a conclusion should be approached with caution. Namely, considering an incremental and not a cumulative model of the influence of risk factors on anxiety [92], the influence and contribution of different risk factors to the level of anxiety can be different. Considering the impact of the COVID-19 pandemic on the population’s mental health, the dominant risk factor affecting anxiety can be assumed. Under that assumption, the obtained result could not be interpreted as supporting the theoretical position of unidimensionality because it is predominantly associated with one specific threatening factor and one threatening situation. The generalization of other threatening situations can be misleading.

### 4.3. GLMM Analysis

It can be noticed that the influence of individual independent factors of interaction terms on STAI-S and STAI-T is different. It can be seen that the pattern of interaction is different in all four mothers’ education level groups. For the STAI-S, there is an interaction between the *number of children in the family and the presence/absence of disorders* in the mothers’ education level groups up to 12 years and from 14 to 16 years, while there is no interaction between these two factors for the other two mothers’ education groups (from 12 to 14 years and more than 16 years). The situation is different for STAI-T, as there exists an interaction between *the number of children in the family and the presence/absence of disorders* independently from the mothers’ education level. One-way ANOVA revealed a statistically significant difference between the subgroups of the grouping variable *number of children in the family* * *presence/absence of disorders* in the different groups based on mothers’ education level. For example, in the mothers’ education level group of *up to 12 years*, we found a statistically significant difference for STAI-T between mothers with one child without disorders (STAI-T_Mean_ = 52) and one child with a disorder (STAI-T_Mean_ = 38.33); one child without disorders and two or more children without disorders (STSAI-T_Mean_ = 39); two or more children where one has a disorder (STAI-T_Mean_ = 50.11) and one child with a disorder; and two or more children with a child having a disorder and two or more children without disorders. In the mothers’ education level group *from 12 to 14 years*, we found a statistically significant difference for STAI-T between mothers of two or more children without disorders (STAI-T_Mean_ = 46.76) and one child with a disorder (STAI-T_Mean_ = 34.29); and two or more children without disorders and two or more children where one has a disorder (STAI-T_Mean_ = 42.87). For the mothers’ education level group *from 12 to 14 years*, we found a statistically significant difference for STAI-S and STAI-T between the observed subgroups. The distribution of the STAI-T mean values for the statistically significant subgroups in different groups of mothers’ education level is different. While mothers of one child without disorders have the highest level of STAI-T in the group with up to 12 years of education, at the same time, the equivalent subgroup of mothers have the lowest level of STAI-T in the group from 14 to 16 years of education. In the studies conducted during 2021, different results have been obtained from research on the influence of the number of children in a family on parenting stress [40,93]. In one study [40], parents with one child have a higher stress level, while in the other study [93], parents of two children have a higher stress level. A possible reason for these contradictory results is different cultural backgrounds and life expectations. It implies that adverse conditions can bring different outcomes, so the level of education can lead to the results we obtain. 

Research that considered the levels of anxiety and stress of the parents of children with disorders showed increased levels of anxiety and stress during the pandemic [4,40,63,67]. Similar studies before the COVID-19 pandemic also showed higher anxiety levels in parents having children with disabilities [63,78,94,95]. Our results do not align with those results because they did not reveal straightforward causality between anxiety and the children’s disorders. We found complex interactions between education level, the number of children, and the presence/absence of disorders. There can be more reasons, from differences in sample characteristics to differences in cultural background, including different statistical analysis approaches. The obtained results can be observed in the light of cumulative and multiple risk models [92] for anxiety prediction. The cumulative model implies that the level of anxiety is influenced by the number of risk factors, and the greater the number of factors, the higher the level of anxiety.

On the other hand, according to the multiple risk model, the severity and variability of risk factors can predict anxiety incrementally, allowing for the specific and relative influence of a risk factor on anxiety. In the observed sample, the risk factors are related to the COVID-19 pandemic, the responsibility of raising children, and the burden of raising a child with a disorder. According to both models, it could be expected that there is a trend to the number of risk factors or the number and weight of individual factors in the observed population of mothers. The results do not indicate such a dependence of anxiety (STAI-T) on the observed factors. The GLMM model indicates that there is an interdependence of demographic and risk factors, indicating the necessity of including complex interactions in the modeling of anxiety.

Additional research should be conducted to resolve those ambiguities.

## 5. Conclusions

The mental health of mothers analyzed in relation to elevated levels of anxiety, a typical psychological response during the COVID-19 pandemic, was suboptimal. This study highlights elevated self-reported anxiety levels in mothers with and without neurodevelopmental disorders during the COVID-19 lockdown. There is no doubt that there is a connection between the pandemic and the increased levels of anxiety among mothers, regardless of whether they have a child with a disorder. Our research reveals that interactions rather than independent variables significantly impact anxiety. Such findings impose a need for psychological support for mothers and, generally, parents, which may positively affect their psychological well-being and offspring development. Also, further research, which should address long-term outcomes, is needed to additionally understand the effects of the pandemic on the parents of children with and without neurodevelopmental disorders. This may prompt changes in government strategies for future pandemic-related crises to maintain children’s and parents’ mental health at optimal levels. 

### Strengths and Limitations

This study is the first in Serbia that considered the sensitive population of parents with and without developmental disabilities, alongside their personal feelings, during the COVID-19 lockdown. It presents a unique opportunity to reveal the heavy burden on parents during the pandemic due to strictly implemented measures.

The study collected cross-sectional data and obtained results from mothers. In that sense, the sample does not represent the population of both parents. We did not have the opportunity to overcome this gap because we conducted the research during the first pandemic wave. We could not supplement the sample in later research with the aim of including fathers because the data would need to be sufficiently comparable. Also, the questionnaire was distributed through electronic communication via social media and online networks, which implies that only highly motivated parents using these communication channels participated in the study. Accordingly, the study results may have projected more of these types of families. Secondly, our measures are based on self-report, so a misunderstanding of questions or social prejudice in parents can violate the validity of findings. Thirdly, the parents who were more concerned regarding COVID-19 were more likely to complete the questionnaire, which can affect the generalizability of the findings. 

## Figures and Tables

**Figure 1 children-10-01292-f001:**
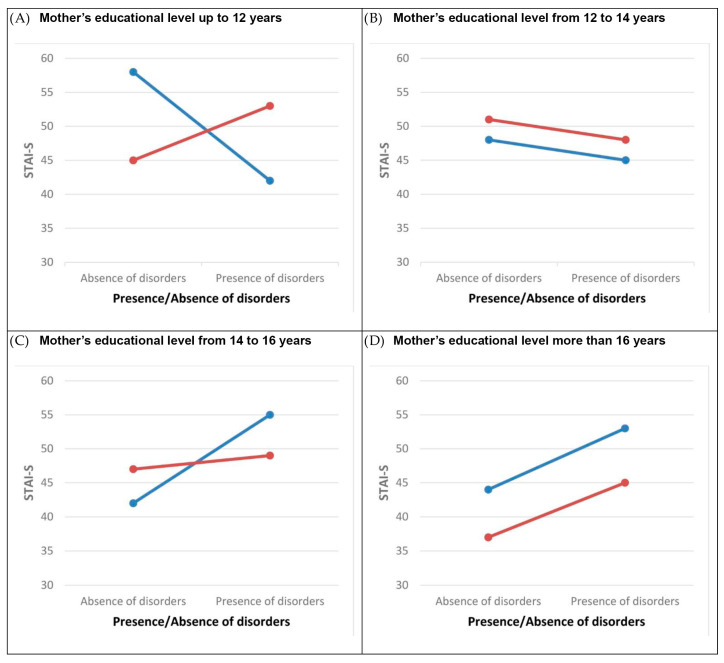
The plot of the STAI-S-estimated marginal means for interactions: *mothers’ education level * number of children in the family * presence/absence of disorders*. Legend: 
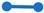
, One child in the family; 
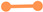
, two or more children in the family. Note: Marginal means are estimated for the maternal age = 38.35. Abbreviations: STAI-S, state anxiety.

**Figure 2 children-10-01292-f002:**
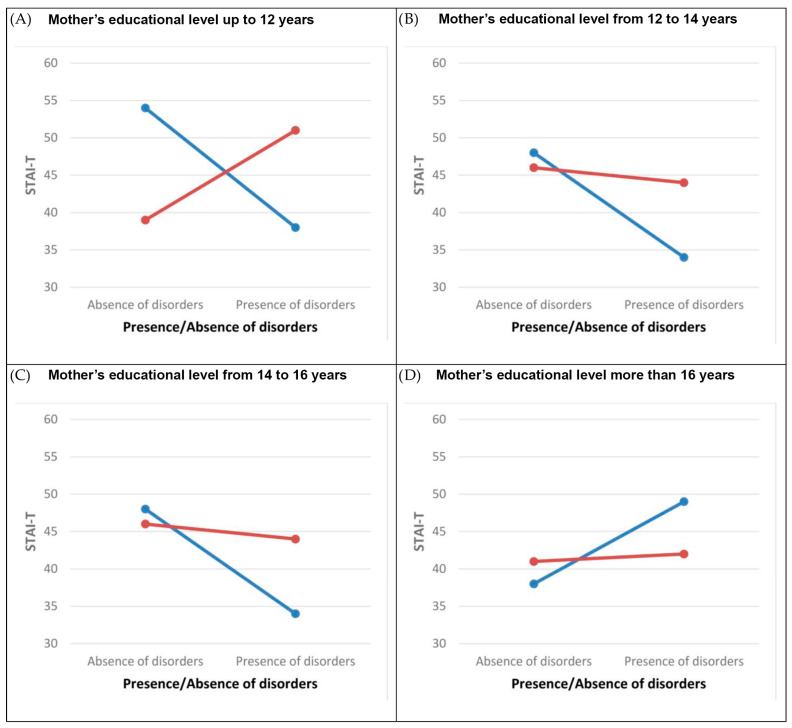
The plot of the STAI-T-estimated marginal means for interactions: *mothers’ education level * number of children in the family * presence/absence of disorders*. Legend: 
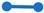
, One child in the family; 
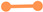
, two or more children in the family. Note: Marginal means are estimated for the maternal age = 38.35. Abbreviations: STAI-T, trait anxiety.

**Figure 3 children-10-01292-f003:**
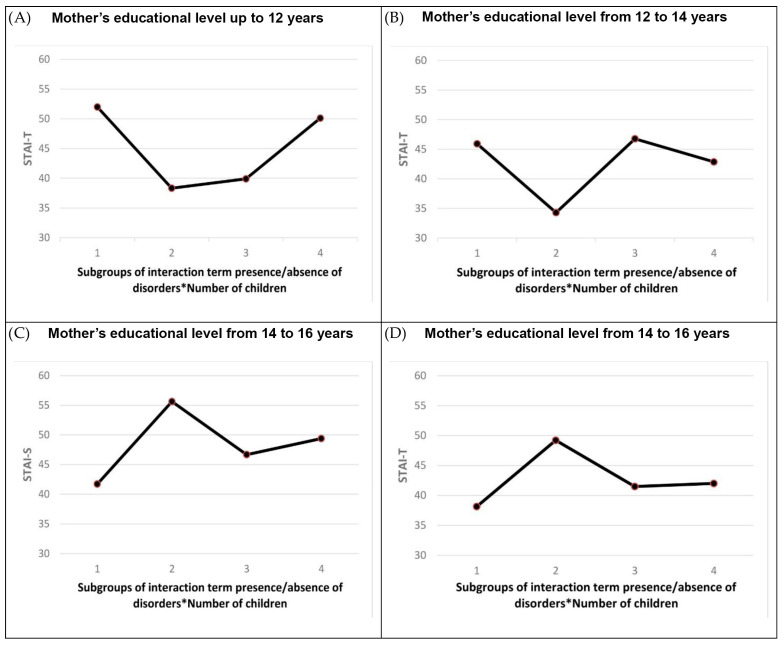
The plot of the STAI-S and STAI-T mean values for different groups of mothers’ education level. Abbreviations: STAI-S, state anxiety; STAI-T, trait anxiety.

**Table 1 children-10-01292-t001:** Sample characteristics (N = 280).

Sample Characteristics	Mothers (N = 280)	
	Number	Percentage %
**Age**		
20–30	23	(8.2)
31–40	171	(61.1)
41–50	80	(28.6)
51 and above	6	(2.1)
**Mothers’ education level**		
Up to 12 years	67	(23.9)
From 12 to 14 years	50	(17.9)
From 14 to 16 years	131	(46.8)
More than 16 years (Ph.D. and higher degrees)	32	(11.4)
**Number of children in the family**		
One child	95	(33.9)
Two and more children	185	(66.1)
**Presence/absence of disorders**		
Yes	122	(43.6)
No	158	(56.4)
**Type of neurodevelopmental disorders**		
Specific language impairment	40	(32.8)
Autism spectrum disorder	82	(67.2)
**STAI-S**		
Low	23	(8.2)
Intermediate	88	(31.4)
High	169	(60.4)
**STAI-T**		
Low	28	(10.0)
Intermediate	144	(51.4)
High	108	(38.6)

Note: STAI-S, state anxiety; STAI-T, trait anxiety.

**Table 2 children-10-01292-t002:** Descriptive statistics on STAI-S and STAI-T.

Variable	Min.	Max.	Mean	SE	SD
**STAI-S**					
Low	23	30	26.87	0.52	2.51
Intermediate	31	45	38.35	0.42	3.95
High	46	72	56.02	0.52	6.78
In the whole sample	23	72	46.69	0.70	11.75
**STAI-T**					
Low	23	30	27.68	0.42	2.23
Intermediate	31	45	37.79	0.36	4.33
High	46	70	54.03	0.63	6.59
In the whole sample	23	70	43.04	0.63	10.55

Note: STAI-S, state anxiety; STAI-T, trait anxiety.

**Table 3 children-10-01292-t003:** Correlations analysis of maternal age, STAI-S and STAI-T.

		Age		STAI-S		STAI-T	
		r	*p*	r	*p*	r	*p*
**Total**	**Age**	1		0.092	0.123	0.046	0.448
	**STAI-S**	0.092	0.123	1		**0.802 ****	**0.001**
	**STAI-T**	0.046	0.448	0.802 **	0.001	1	
**Mothers’ education level up to 12 years**	**Age**	1		−0.141	0.256	−0.222	0.071
	**STAI-S**	−0.141	0.256	1		**0.766 ****	**0.001**
	**STAI-T**	−0.222	0.071	**0.766 ****	**0.001**	1	
**Mothers’ education level from 12 to 14 years**	**Age**	1		0.134	0.356	0.037	0.8
	**STAI-S**	0.134	0.356	1		**0.821 ****	**0.001**
	**STAI-T**	0.037	0.8	**0.821 ****	**0.001**	1	
**Mothers’ education level from 14 to 16 years**	**Age**	1		**0.203 ***	**0.02**	**0.232 ****	**0.008**
	**STAI-S**	**0.203 ***	**0.02**	1		**0.795 ****	**0.001**
	**STAI-T**	**0.232 ****	**0.008**	**0.795 ****	**0.001**	1	
**Mothers’ education level more than 16 years**	**Age**	1		0.076	0.681	0.048	0.793
	**STAI-S**	0.076	0.681	1		**0.860 ****	**0.001**
	**STAI-T**	0.048	0.793	**0.860 ****	**0.001**	1	

Note: *, Correlation is significant at the 0.05 level (2-tailed); **, Correlation is significant at the 0.01 level (2-tailed). STAI-S, state anxiety; STAI-T, trait anxiety.

**Table 4 children-10-01292-t004:** GLMM model results for STAI-S.

Source	F	df1	df2	Sig.
**Corrected model**	2.974	16	263	0.000
**Maternal age**	5.860	1	263	0.016
**Mothers’ education level * presence/absence of disorders**	4.974	3	263	0.002
**Mother’ education level * number of children in the family * presence/absence of disorders**	6.112	3	263	0.000

Note: Probability distribution: normal; link function: identity; Sig.: statistical significance.

**Table 5 children-10-01292-t005:** GLMM model results for STAI-T.

Source	F	df1	df2	Sig.
**Corrected model**	4.200	16	263	0.000
**Mothers’ education level * presence/absence of disorders**	6.884	3	263	0.000
**Number of children in the family * presence/absence of disorders**	5.224	1	263	0.023
**Mothers’ education level * number of children in the family * presence/absence of disorders**	10.202	3	263	0.000

Note: Probability distribution: normal; link function: identity; Sig.: statistical significance.

**Table 6 children-10-01292-t006:** Explanation of new grouping variable used in one-way ANOVA statistical test.

Number of Children	Presence/Absence of Disorders	Subgroup
1	No	1
1	Yes	2
>1	No	3
>1	Yes	4

## Data Availability

Not applicable.

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
