# Peer review of "Comparing Anxiety Levels during the COVID-19 Pandemic among Mothers of Children with and without Neurodevelopmental Disorders"

_children, 2023, doi:10.3390/children10081292_

Round 1

Reviewer 1 Report

Dear authors, thank you very much for the opportunity to review this manuscript. The topic is extremely important. I have a few comments:

-The introduction lacked explanation of the essence of ASD and SLI disorders (symptoms, characteristic behaviors of children with these disorders);

- I would also suggest adding a few words about how the care of these children works.

- In my opinion, the discussion lacks an explanation that the increased anxiety could result from the fact that the rehabilitation of children was often interrupted due to the closure of the institution by Covid-19

- and it is well known how important systematic rehabilitation is

Author Response

Dear Reviewer, 

Please find the enclosed revised version of the manuscript. 

Thank you very much for your suggestions. 

Kind regards

Reviewer 2 Report

The paper discusses the differences in COVID-19-related anxiety levels between mothers of children with and without neurodevelopmental disorders. In the introduction, the authors briefly review the COVID-19 pandemic and some literature about related anxiety found in families during that time. While data is available regarding general anxiety levels, this paper discusses a more specific population, as previously mentioned.

Following a screening process, data has been collected and analyzed. Beyond the expected generally elevated anxiety, the results show a significant correlation between the state and trait anxiety scale using the State-Trait Anxiety Inventory (STAI). Moreover, a number of interactions were found between the age of the mothers, their education and their children (with or without neurodevelopmental disorders).

A few comments about the paper:

·         Figure 1, 2 and 3 – labeling each square (A, B C and D) with “mother’s education” is advisable.

·         It is not clear what does a correlation between STAI-S and STAI-T mean. Adding a brief explanation, ideally with references, would be beneficial.

·         GLMM analysis – it would be beneficial to elaborate on the findings, as opposed to simply mention the observed significant difference and interactions. What was the difference? More or less anxiety and for who? Some of the simple description should be moved to the “results” section.

·         Lacking discussion regarding a potential explanation for the findings. What could all that mean, based on existing literature?

Author Response

Dear reviewer,

Please find enclosed the revised version of the manuscript.

Thank you for your comments and suggestions!

Kind regards
